# Soil Sensor Use in Delimiting Management Zones for Sowing Maize in No-Till

**DOI:** 10.3390/s24237552

**Published:** 2024-11-26

**Authors:** Eduardo Leonel Bottega, Ederson Bitencourt Pinto, Ezequiel Saretta, Zanandra Boff de Oliveira, Filipe Silveira Severo, Johan Assmann

**Affiliations:** Academic Coordination, Campus Cachoeira do Sul, Federal University of Santa Maria, Santa Maria 97105-900, Brazil; eder-bitencourtpinto@hotmail.com (E.B.P.); ezequiel.saretta@ufsm.br (E.S.); zanandra.oliveira@ufsm.br (Z.B.d.O.); filipesevero2011@hotmail.com (F.S.S.); johanassmann1@gmail.com (J.A.)

**Keywords:** *Zea mays*, precision agriculture, apparent soil electrical conductivity

## Abstract

This study aimed to analyze yield components and maize yield cultivated at different population densities in management zones (MZs) delimited based on mapping the spatial variability of the soil’s apparent electrical conductivity (ECa). The soil ECa was measured, and two MZs were subsequently delimited, one with low ECa and the other with high ECa. In each MZ, four maize sowing densities were tested: 60,000 (D1); 80,000 (D2); 100,000 (D3); and 140,000 (D4) seeds ha^−1^. Ear length, number of grains per ear, number of grains per row, number of rows per ear, thousand-grain weight, and yield were evaluated. The increase in sowing density in the high ECa MZ linearly reduced the values of ear diameter, number of rows per ear, number of grains per ear, and thousand-grain weight. Sowing density D3, when implemented in the low ECa MZ, showed higher values for the ear length, ear diameter, number of grains per row, number of grains per ear, and thousand-grain weight. Sowing density D2 was the one with the highest yield, regardless of the MZ where it was implemented (5628.48 kg ha^−1^ in the high ECa management zone and 4463.63 kg ha^−1^ in the low ECa).

## 1. Introduction

Maize (*Zea mays* L.) is one of the most significant crops globally. It is grown over a wide geographical area and under various climatic and soil conditions. To maximize the genetic potential of the available varieties and hybrids, it is essential that they are cultivated according to best agricultural practices and in suitable environments for crop management [1].

Maize production in Brazil for the 2023/24 season was approximately 115.6 million tons [2]. The 2023/24 harvest presented a total cultivated area estimated at 20.9 million hectares, reducing the area cultivated in the previous harvest by 5.9%. In most of the states, there was a reduction in productivity compared to the last harvest due to climate instability. One of the exceptions was Rio Grande do Sul, which showed a significant recovery in productivity, where there was a 2.2% increase [2]. 

The use of different maize cultivars associated with strategies aimed at increasing crop yield is thus becoming a focus among studies in the scientific environment. Plant density is one of the most important cultural practices determining grain yield, as well as other important agronomic attributes of maize [3]. Plant density can vary widely and is related to several factors, such as the duration of the hybrid’s growing season, the morphological characteristics of the plant, its growth habit, the amount and distribution of rainfall during the growing period, soil moisture reserves during winter, soil fertility levels, the timing of planting, crop management, and biomass, as well as the final yield [4].

The use of technologies applied to management in rural areas considering the variability of soils and crop characteristics in space and time is crucial for better interpretation and seeking better control in decision-making for agricultural crops. Precision agriculture (PA) provides adequate tools for this type of management. In recent years, precision agriculture has emerged as a practice that divides individual fields into management zones (MZs) according to the similarities in soil characteristics [5]. 

Soil exhibits significant spatiotemporal variability influenced by a range of internal and external factors. A highly effective strategy for managing this spatial variability is the implementation of management zones (MZs) [6]. The spatial variability mapping of soil’s apparent electrical conductivity has been studied as a potential tool for defining these management zones [7,8]. The chemical and physical attributes of the soil that cause variations in its apparent electrical conductivity are also responsible for variations in crop productivity [9]. 

The present study aimed to analyze yield components and maize yield cultivated at different sowing densities in delimited management zones based on the spatial variability mapping of the soil’s apparent electrical conductivity.

## 2. Materials and Methods

The experiment was conducted from September 2021 to February 2022 in Vila Piquiri in the municipality of Cachoeira do Sul-RS, with coordinates 30°13′49″ south latitude and 52°47′17″ west longitude, in an area destined for the cultivation of grains, in a plot with 17.2 ha. The predominant climate in the region is classified as humid subtropical [10]. The soil in the region is classified as Argisol [11]. In the year 2020, during the summer, the direct planting of soybeans was carried out. Then, during the winter, harrowing was carried out, and millet was sown (pasture). Subsequently, after the millet harvest, the maize used in the experiment was sown.

Initially, sampling grids with 89 points regularly spaced in 50 × 50 m were established (Figure 1). These points served as the basis for measuring the apparent electrical conductivity of the soil (ECa, mS m^−1^), a soil attribute that was later used to delimit the management zones (MZs). A Garmin GPS receiver, model GPSMAP 62sc, was used to locate the sample points in the study areas.

ECa was measured via the electrical resistivity method. Four electrodes equally and horizontally spaced at 0.20 m were introduced into the soil with the aim of measuring the representative ECa of the soil layer from 0.0 to 0.20 m. The configuration and assembly of the electrodes was based on the Wenner matrix [12,13].
(1)$$\rho =\frac{2\pi a\Delta V}{i}$$
where ρ = resistivity (Ohm m^−1^); a = spacing between electrodes (m); Δ*V* = measured potential difference (V); and *i* = applied electric current (A).

The apparent soil electrical conductivity represents the inverse of resistivity and is calculated using Equation (2).
(2)$$ECa=\frac{1}{\rho }$$
where *ECa* = apparent soil electrical conductivity (mS m^−1^).

To obtain the soil’s apparent electrical conductivity, a portable electrical conductivity meter, Landviser^®^, model LandMapper^®^ ERM-02, was used. The meter has the ability to measure the difference in potential (voltage) between the points, making it possible to obtain the electrical resistance of water (humidity) in the soil. The Wenner matrix consisted of a “T” shaped frame, with metalon–metal tubes, steel screws (electrodes), and flexible wires of different colors. The red wires were connected to the current electrodes, and the black wires were connected to the potential electrodes. To ensure reliability in the readings, the contact of the screws with the metalon–metal frame was properly insulated, and they were coated with PVC hose. This type of portable sensor is easy to operate. Its use may present limitations in stony soils, as they have a greater amount of air inside, making it difficult to conduct electrical currents. Figure 2 shows the Wenner Matrix and ECa meter (Figure 2a) and Electrical Resistivity Meter Landviser^®^, model LandMapper^®^ ERM-02 (Figure 2b).

At the time of ECa readings, soil samples (0.0–0.20 cm) were taken to determine soil moisture. The samples were collected using a Dutch auger, stored and protected in an aluminum capsule, duly identified, and taken to the laboratory. To estimate soil moisture, the gravimetric method (greenhouse standard) was used. The samples were weighed while still wet, placed in an oven at 105 °C for 24 h, and weighed after they dried. Both the wet weight and the dry weight were deducted from the aluminum capsule weight.

After taking the ECa readings, the spatial dependence of this soil attribute was modeled, and the management zones were delimited. The spatial dependence of the apparent electrical conductivity of the soil was evaluated using the model and validated by the cross-validation technique. After adjusting and validating the model, the interpolation of values for prediction in non-sampled locations was performed. The interpolation method adopted was ordinary kriging because it provides the best unbiased linear forecasts [14].

The analysis of spatial variability and the delimitation of management zones was carried out in the Geographic Information System (GIS) Quantum Gis, version 3.10.11 (A Coruña, Spain), using the Smart-Map plugin [15]. Smart-Map enables the prediction and mapping of soil attributes. It allows the interpolation of data using ordinary kriging and Machine Learning techniques through the Support Vector Machine (SVM), as well as the delimitation of management zones.

After delimiting the MZ, maize sowing was carried out. For sowing, a Ford 4 × 2 TDA tractor, model 7610, with 103 hp power, and a fertilizer seeder, Semeato brand, model PSE 8, with 6 rows spaced at 0.5 m, were used. Maize was sown on 28 September 2021, following the agroclimatic zoning of maize in the region. After the maize was sown, 11.2 mm of rain fell in the area. The sowed hybrid was the LG RNC 3040VIP3. This hybrid has Agrisure Viptera 3^®^ protection benefits, with a recommended sowing density of plants equivalent to 60,000 plants per hectare. In this study, the following sowing densities were tested: 60,000 (1), 80,000 (2), 100,000 (3), and 140,000 (4) seeds per hectare. This aimed to model the productive behavior of the implanted hybrid based on the regression analysis.

The area used for sowing the different sowing densities studied was equivalent to 66 m^2^ in each MZ, totaling 132 m^2^. The experiment was set up following the statistical design of randomized blocks (2 × 4) with 3 replications, 2 MZs (high and low ECa), and 4 sowing densities. The plots consisted of 3 sowing lines 10 m in length.

The following variables were evaluated for each sowing density of plants in each MZ: ear length (EL, cm), ear diameter (ED, cm), number of rows per ear (NRE), number of grains per row (NGR), number of grains per ear (NGE), thousand-grain weight (GW, g), and yield (YLD, kg ha^−1^). The ears present in the plants in 3 rows 10 m in length were manually harvested in each repetition of each treatment. EL and ED were obtained by measuring the ears with a graduated ruler; NRE, NGR, and NGE were obtained by manual counting; GW was obtained by weighing a thousand grains on a precision scale; and YLD was estimated by extrapolating the obtained weight by area of each repetition for 1 hectare.

The obtained values were recorded in an Excel spreadsheet. They were submitted to descriptive statistical analysis, calculating the minimum value, mean, maximum value, standard deviation, and coefficient of variation. A regression analysis was carried out in order to adjust the representative models of each studied component behavior as a function of sowing density and management zone where the crop was implanted. The analyses were carried out in the SISVAR computer program [16].

## 3. Results

Figure 3 shows a graph referring to the rainfall recorded for the period in which the study was carried out. The rainfall recorded during the corn crop cycle was 183 mm.

Figure 4 shows a semivariogram (Figure 4a), thematic maps representing the spatial variability of ECa in the study area (Figure 4b), and the management zones indicating the sowing sites of the different sowing densities studied (Figure 4c). In this study, two well-defined MZs were adopted. Class 1 (red) represents the high ECa zone, and Class 2 (green) represents the low ECa zone.

Table 1 presents the descriptive statistical parameters of the studied variables: apparent soil electrical conductivity (ECa; mS m^−1^), soil moisture (SM, g g^−1^), ear length (EL, cm), ear diameter (ED, cm), number of rows per ear (NRE), number of grains per row (NGR), number of grains per ear (NGE), thousand-grain weight (GW, g), and productivity (YLD, kg ha^−1^). We observed the highest mean number of grains per ear (530.19) in MZ 1 (high ECa), which resulted in higher maize mean productivity equivalent to 3904.24 kg ha^−1^. The number of grains per row has a negative indirect effect on grain yield; therefore, a reduction in this income component will cause a loss in yield [17], as observed in this study.

The adjusted models based on the regression analysis for the analyzed variables are presented in Figure 5, namely, ear length (Figure 5a), ear diameter (Figure 5b), number of rows per ear (Figure 5c), number of grains per row (Figure 5d), number of grains per ear (Figure 5e), and thousand-grain weight (Figure 5f), observed for the different sowing densities as a function of the management zones where they were implanted (MZ 1: high ECa; MZ 2: low ECa). The quadratic function was the function that best adjusted to the observed results for the evaluated variables when maize was sown in MZ 2, regardless of the sowing density studied. For the sowing densities implanted in MZ 1, the linear function was the function that best adjusted to the results observed for the ear diameter, number of rows per ear, number of grains per ear, and thousand-grain weight variables, showing that the increase in sowing density causes a reduction in the values of these variables. For the observed values of the ear length and number of grains per row variables, the quadratic model was the best fit. The best adjustments (higher R^2^ values) were obtained for the different sowing densities in MZ 1, with 0.89 being the lowest and 0.99 being the highest R^2^ value. As for the model adjustments of the analyzed variables in MZ 2, they presented lower R^2^ values, with 0.58 being the lowest and 0.98 being the highest value.

Regarding the ear length, the highest sowing density caused a reduction in this yield component, regardless of the MZ where it was implanted. For sowing in MZ 1, densities 1, 2, and 3 did not show differences, whereas, in MZ 2, the greatest ear length (16 cm) was obtained for density 3 (100,000 seeds per hectare). Each hybrid has its specific characteristics of ear length, which may vary due to different environmental conditions during the agricultural year.

Sowing density 3 (100,000 seeds ha^−1^) in the low ECa zone (MZ 2) showed, on average, the highest number of grains per ear, equivalent to 623 grains. The high ECa zone (MZ 1) had the highest mean number of grains per ear, with 581 grains at sowing density 1 (60,000 seeds ha^−1^).

Sowing density 4 (140,000 seeds ha^−1^) in the low ECa zone showed, on average, the highest number of rows per ear, equivalent to 18.8 rows. The high ECa zone, on the other hand, had the highest number of grains per row at sowing density 1 (60,000 seeds ha^−1^), with a mean value of 18.6 rows (Figure 5c).

Sowing density 3 (100,000 seeds ha^−1^) in the low ECa zone showed, on average, the highest number of grains per row, equivalent to 35 grains. The high ECa zone, on the other hand, had the highest number of grains per row at sowing density 1 (60,000 seeds ha^−1^), with a mean value of 31 grains (Figure 5d).

Regarding the thousand-grain weight (GW), sowing density 3 (100,000 seeds ha^−1^) in the low ECa zone presented, on average, the highest value, equivalent to 136.30 g. The high ECa zone, on the other hand, presented the highest GW for sowing density 1 (60,000 seeds ha^−1^), with a mean value of 129.03 g.

Figure 6 presents the adjusted models based on the regression analysis for the maize yield (kg ha^−1^) observed for the different seeding rates studied and implemented in the different management zones. Both in the high ECa MZ (MZ 1) and the low ECa MZ (MZ 2), sowing density 2 (80,000 seeds ha^−1^) had the highest yield.

In MZ 1, the obtained yield was 5628.48 kg ha^−1^, and in MZ 2, it was equivalent to 4463.63 kg ha^−1^. These yields were, respectively, 31 and 50.7% higher than those obtained for the recommended sowing density of the cultivated hybrid (60,000 seeds ha^−1^).

## 4. Discussion

Extreme weather conditions are one of the main causes of low corn yield [18]. Due to the atypical year caused by the ‘‘La Niña” event during the experiment period, the drought recorded in the spring–summer of 2021/2022 in the state of Rio Grande do Sul had a negative impact on the maize grain yield (−53% in productivity and −55% in production). When analyzing the first quarter of 2022 (January–February–March), we verified that in this period, there was low volume and irregularity in river precipitation, especially during the last week of February [19]. In the state of Rio Grande do Sul, the chances of above-normal rainfall are greater in El Niño years and below-normal rainfall in La Niña years, influencing the climate rhythm of Rio Grande do Sul, acting under the rhythm of front displacement [20]. 

The maize hybrid in the Brazilian territory has a varied water demand, which depends on its climatic condition; thus, a 380 to 550 mm water demand can be required [21]. The maize crop may present a drop in production due to the occurrence of water stress periods, since the lack of water affects crop development, especially between the flowering stage and physiological maturation [22]. Water deficit in the moment before the anthers release can reduce the grain yield by half; if it occurs in full flowering, it can cause a 20% to 50% drop in a period of 2 to 8 days, respectively [23]. Grain productivity is more affected if water stress occurs in the pollination, zygote formation, and grain-filling phases [24].

It has already been proven that soil conducts electricity in three ways via (1) the liquid phase, through the amount of water and dissolved solids in this phase that are present in the macropores of the soil, (2) the solid–liquid phase, depending on the concentration of exchangeable cations associated with clay minerals, and (3) the solid phase, based on the proximity between soil particles [25]. Considering that soil presents spatial variability in its water storage capacity, depending on the existing variation in its physical attributes, and that variations in these attributes influence the electrical behavior of soil, mapping this characteristic becomes strategic in management decision-making. 

Two very distinct regions were mapped, one of high and one of low electrical conductivity (Figure 4). The high ECa region is the result of the large difference in levels, as this site is close to an irrigated rice crop and, therefore, has higher humidity, given the low elevation of the site. The low ECa region is the result of the management adopted on the property (conventional cultivation), which reduces the content of organic matter and nutrients [26], thus modifying the soil physical characteristics that directly influence the storage of soil water, essential for the conduction of electric current.

Variations in ECa values are explained by variations in soil attributes, whether chemical or physical. Positive correlations between cation exchange capacity, organic matter (OM) content, pH, H_2_O, calcium, magnesium, and the sum of bases, and a negative correlation with aluminum were observed in different studies [27,28], demonstrating that the mapping of ECa constitutes a tool sensitive to attributes conditioned by soil acidity. These same attributes are capable of significantly interfering with the development of maize and its yield [29], as observed in Figure 5.

When studying the correlation of soil attributes with ECa in different soils (sandy and clayey), researchers observed that in the sandy soil area, there was a positive correlation with the remaining phosphorus, and in the clayey soil area, the correlation between ECa and the content of clay was significantly positive but negative for the sand content [30]. In another study, it was observed that maize grain productivity was related to ECa, indicating that this soil attribute can be used to delimit management zones that reflect the different grain productivity potentials, thus enabling the management of soil attributes associated with the yield performance of crops [31].

The ECa variations allowed the delimitation of the study area into two management zones (Figure 4), which is considered an adequate number as it facilitates management due to the smaller number of distinct regions to be worked. The use of a large number of classes creates small management zones, which tend to increase the irregularity of the areas, thus making them difficult to manage [32].

The management of the sowing densities, based on management zone delimitation, can result in yield gains because the sowing densities have a direct influence on the yield components of the crop, as observed in this study (Figure 5). The reduction in the sowing density of plants caused an increase in the ear length [33]. The authors point out that this behavior may be associated with low plant density, which reduces competition for nutrients, allowing the maximization of post-anthesis photosynthetic activity.

In Rio Grande do Sul (RS), the probability of water deficiency occurring between December and January is high and recurrent, resulting in yield losses due to irregular water distribution and availability throughout the phenological cycle of maize [34]. In this context, mapping the spatial variability of soil’s apparent electrical conductivity (ECa) becomes crucial, as it enables the identification of production field locations with a greater potential for water storage. This, in turn, aids in sowing management and the selection of hybrids to be seeded. 

The increase in the sowing densities per unit area by raising the sowing density is an efficient way to increase the interception of incident solar radiation in the crop. However, the use of very high densities can negatively affect the crop’s photosynthetic activity and the efficiency of converting photoassimilates into grain production. This results in an increase in female sterility, as well as a reduction in the number of grains per ear and grain yield [35].

Increased maize yield has been observed when sown at higher densities and under conditions of greater water supply [36]; however, in the absence of water stress, maize yield is not significantly affected by increased plant population [34]. This study found differences in maize yield based on the management zone (MZ) in which it was sown. This information can contribute to the adoption of population management strategies aimed at increasing yield in areas where it is lower. A study examining the interactions among genotype (G), environment (E), and management (M) in the arid northwest of China revealed that maize yield can be enhanced by at least 10%. This suggests that efforts to increase food production should be concentrated in low-yielding zones [37].

The number of grains per row (NGR) is directly related to the mean ear length, and its potential depends on the interaction between the hybrid and the environment. The lower the NGR, the lower the number of grains per ear; thus, it is detrimental to the productive potential of the hybrid [38,39].

A study demonstrated that in management zones with higher yield, the increase in sowing density had less influence on the number of grains per ear and grain weight. Therefore, in low yield management zones, the thousand-grain weight, considering a 46,000 plants ha^−1^ sowing density, presented approximately 15 g for each additional 10,000 plants; consequently, the number of grains per ear reduced with the increase in seed density sowing; therefore, no increase in the number of grains per m^2^ was observed [40].

The use of the seed variation rate in Rio Grande do Sul, Brazil, is efficient in increasing yield and profitability in the maize crop [41]. In this study, we demonstrate that for the low yield management zone, a better result was obtained for low plant sowing density; thus, using 46 and 50 thousand plants ha^−1^ in high yield management zones, they obtained better gains, with a higher sowing density of 77 and 82 thousand plants ha^−1^.

## 5. Conclusions

The apparent soil electrical conductivity showed spatial variability in the studied area, allowing the delimitation of two management zones.

The increase in sowing density in the management zone of high apparent soil electrical conductivity linearly reduced the ear diameter, number of rows per ear, number of grains per ear, and thousand-grain weight values.

Sowing density D3, equivalent to 100,000 seeds ha^−1^, when implanted in the management zone of low apparent soil electrical conductivity, showed higher values for the ear length, ear diameter, number of grains per row, number of grains per ear, and thousand-grain weight variables.

Sowing density D2, equivalent to 80,000 seeds ha^−1^, showed the highest maize yield, regardless of the management zone where it was implanted. A yield of 5628.48 kg ha^−1^ was obtained in the high apparent soil electrical conductivity management zone and 4463.63 kg ha^−1^ in the low apparent soil electrical conductivity management zone.

## Figures and Tables

**Figure 1 sensors-24-07552-f001:**
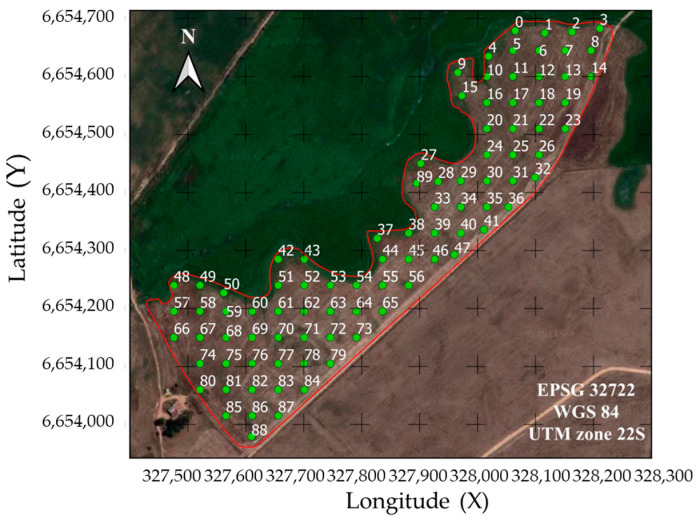
Polygon demarcating the study area and grid of sampling points used as a basis for measuring the apparent soil electrical conductivity (ECa, mS m^−1^).

**Figure 2 sensors-24-07552-f002:**
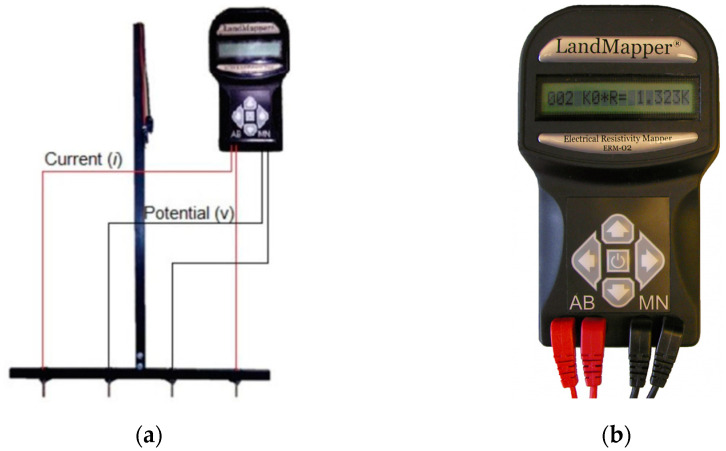
Wenner Matrix and ECa meter (**a**) and Electrical Resistivity Meter Landviser^®^, model LandMapper^®^ ERM-02 (**b**).

**Figure 3 sensors-24-07552-f003:**
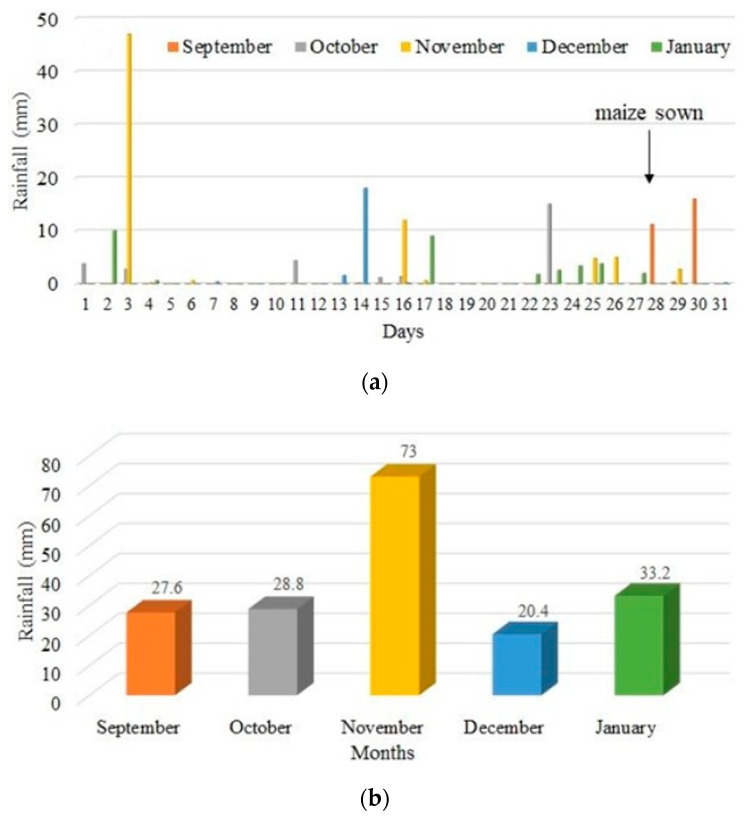
Daily (**a**) and cumulative (**b**) precipitation recorded for the months in which the study was conducted.

**Figure 4 sensors-24-07552-f004:**
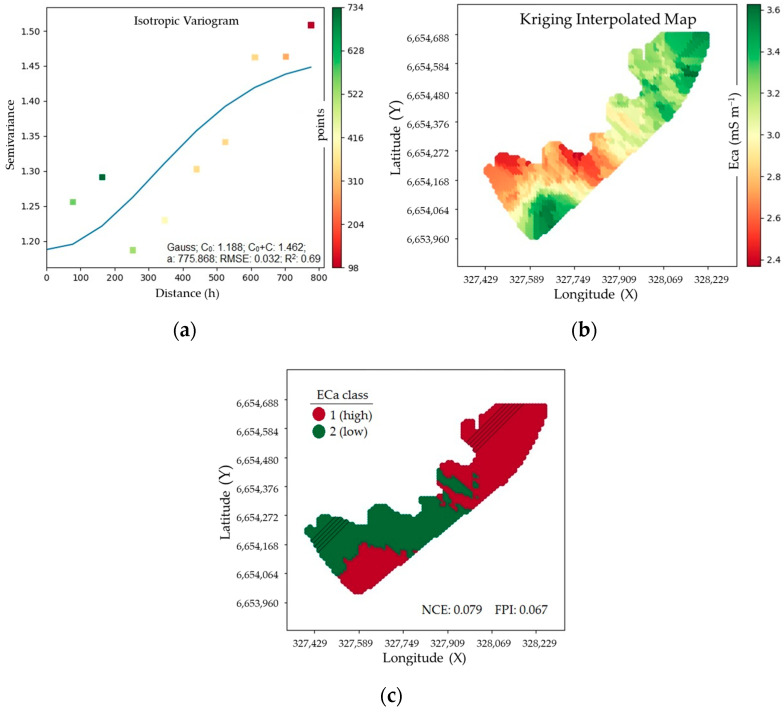
Semivariogram (**a**), thematic maps representing the spatial variability of ECa in the study area (**b**), and management zones (**c**) indicating the sowing sites of the different sowing densities studied.

**Figure 5 sensors-24-07552-f005:**
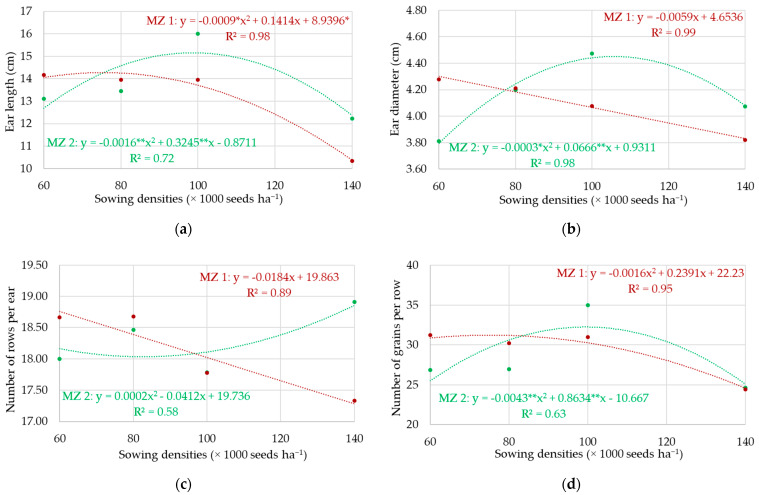
Ear length (**a**), ear diameter (**b**), number of rows per ear (**c**), number of grains per row (**d**), number of grains per ear (**e**), and thousand-grain weight (**f**) observed for the different sowing densities depending on the management zones where they were implanted (MZ 1: high ECa; MZ 2: low ECa). ** Significant coefficient at 1% probability; * significant coefficient at 5% probability.

**Figure 6 sensors-24-07552-f006:**
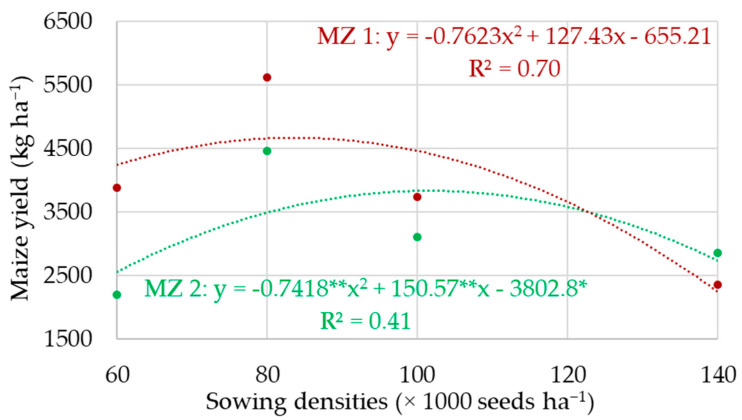
Maize yield observed for the different sowing densities according to the management zones where they were implanted (MZ 1: high ECa; MZ 2: low ECa). ** Significant coefficient at 1% probability; * significant coefficient at 5% probability.

**Table 1 sensors-24-07552-t001:** Descriptive statistics of variables: apparent soil electrical conductivity (ECa; mS m^−1^), soil moisture (SM, g g^−1^), ear length (EL, cm), ear diameter (ED, cm), number of rows per ear (NRE), number of grains per row (NGR), number of grains per ear (NGE), thousand-grain weight (GW, g), and productivity (YLD, kg ha^−1^).

Attribute	Min	µ	Max	σ	VC
Management Zone 1 (high ECa)					
ECa	1.05	3.49	8.60	1.67	47.85
SM	0.19	0.3	0.33	0.01	3.03
EL	9.50	13.10	15.00	1.82	13.93
ED	3.50	4.10	4.80	0.33	8.18
NRE	16.00	18.11	19.33	1.13	6.21
NGR	22.00	29.22	33.67	3.85	13.17
NGE	352.00	530.19	650.90	81.42	15.36
GW	83.20	111.74	162.87	22.32	19.97
YLD	2045.45	3904.24	7456.36	1654.65	42.38
Management Zone 2 (low ECa)					
ECa	1.05	2.87	5.32	1.01	35.19
SM	0.11	0.11	0.12	0.02	18.18
EL	11.83	13.69	17.00	1.60	11.67
ED	3.70	4.14	4.86	0.32	7.71
NRE	17.33	18.29	19.33	0.67	3.69
NGR	22.96	28.36	37.27	4.45	15.7
NGE	429.30	517.71	696.90	75.93	14.67
GW	100.86	121.36	148.32	13.57	11.18
YLD	1794.55	3157.42	4840.00	918.22	29.08

Min = minimum; µ = mean; Max = maximum; σ = standard deviation; VC = variation coefficient (%).

## Data Availability

The raw data supporting the conclusions of this article will be made available by the authors upon request.

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
