# Peer review of "Soil Sensor Use in Delimiting Management Zones for Sowing Maize in No-Till"

_sensors, 2024, doi:10.3390/s24237552_

Round 1
Reviewer 1 Report
Comments and Suggestions for Authors
The study provides detailed data on the impact of different sowing densities within MZs, offering valuable insights for optimizing maize yield. It has clearly demonstrated that the electrical soil conductivity is a useful analytical tool that can provide valuable information with regards to parameters such as planting density, which would result in higher yields and better management practices for agricultural crop production. However, integrating additional soil sensors, such as moisture or nutrient sensors, has become a common practice to refine management zones. The study's exclusive focus on ECa without comparing these additional tools would limit its industrial applicability. Additionally, there is only limited information provided on the how these sensors would be deployed or their ability for frequent and remote monitoring of ECa. I have some concerns about the manuscript content which I have detailed in the comments below. I cannot recommend publication of this article in its present form, but if all comments listed below can be addressed, I may then reconsider.
Comments
1) It has previously been demonstrated that optimal planting densities vary significantly based on soil types and hybrid characteristics The study needs to discuss these dependencies further and assess whether the current findings are universally optimal or specific to their conditions.
2) The current study would require further discussion on how different climatic conditions or annual variations could impact ECa values, which is an important factor in yield prediction.
3) The author should provide further information on how the conductivity sensors would be deployed in a practical sense, can they be read remotely, can the system be easily deployed at multiple sites, can they be utilised in a cost-effective manner, can they be easily integrated with other sensors
4) Table 1 (Between line 172-173) shows that the higher the soil moisture the greater the electrical conductivity, can some further information be provided by the authors as to what other physical parameters of the soil such as porosity could affect electrical conductivity.
Minor points
5) The meaning of PMG (line 138) is not provided in the manuscript
6) The English language in a number of sentences could be improved eg, (The beginning of the experiment occurred by establishing the sampling grid with 89 points regularly spaced in 50 x 50 meters). May be better to be along the lines (Initially sampling grids with 89 points regularly spaced in 50 x 50 meters were established) (lines 67-68)
7) (obtained by measuring the ears with a graduated ruler; NFE, NGF) (line 137). These abbreviations are not correct as not mentioned before.
Author Response
Comments 1: It has previously been demonstrated that optimal planting densities vary significantly based on soil types and hybrid characteristics The study needs to discuss these dependencies further and assess whether the current findings are universally optimal or specific to their conditions.
Response 1: Thanks for pointing this out. We agreed and added the requested information to the paper.
Comments 2: The current study would require further discussion on how different climatic conditions or annual variations could impact ECa values, which is an important factor in yield prediction.
Response 2: Thanks for pointing this out. ECa is influenced by chemical, physical, physicochemical and biological factors in the soil. These factors are isolated and together. The transmission of electrical current in the soil occurs through 3 phases: (1) liquid phase, through the amount of water and solids dissolved in this phase, present in the soil's macropores; (2) solid-liquid phase, dependent on exchangeable cations associated with clay minerals and (3) solid phase, based on the proximity between soil particles (Corwin and Lesch 2005). We understand your point of view on the impact of climatic conditions on grain yield, we know that such conditions also affect soil formation and the activity of its living beings. This discussion required monitoring work data that we did not collect, as the objective was only to identify the spatial variability of ECa in the productive field and use this information to propose sowing corn at different densities as a management strategy. Thank you for the suggestion, which we will take into consideration when carrying out future work.
Corwin, D.L.; Lesch, S.M. Apparent soil electrical conductivity measurements in agriculture. Comput Electron. Agr., v.46, n.1-3, p. 11-43, 2005. DOI: https://doi.org/10.1016/j.compag.2004.10.005
Comments 3: The author should provide further information on how the conductivity sensors would be deployed in a practical sense, can they be read remotely, can the system be easily deployed at multiple sites, can they be utilised in a cost-effective manner, can they be easily integrated with other sensors.
Response 3: Thanks for pointing this out. In the global agricultural market, there is commercial equipment designed to read soil ECa. In the USA, this is Veris (https://www.veristech.com/). In Brazil, two manufacturers have commercial ECa measurement equipment, Stara (https://www.stara.com.br/en/products-services/precision-agriculture/product/veris-ce) and Falker (https: //www.falker.com.br/en/terram). What we seek with our study is to demonstrate the potential for using spatialized information on variations in ECa, in particular, in adjusting the plant population. In Brazil this technology is being under-used, with ECa maps only used to direct soil sampling. The sensors mentioned above operate with the same operating principle as the sensor used in this study, however, they provide a greater work capacity, which is why they are used to provide services in Precision Agriculture.
Comment 4: Table 1 (Between line 172-173) shows that the higher the soil moisture the greater the electrical conductivity, can some further information be provided by the authors as to what other physical parameters of the soil such as porosity could affect electrical conductivity.
Response 4: Thanks for pointing this out. We agreed and added the requested information to the paper.
Minor points
Comment 5: The meaning of PMG (line 138) is not provided in the manuscript.
Response 5: Thanks for pointing this out. We agreed and added the requested information to the paper.
Comment 6: The English language in a number of sentences could be improved eg, (The beginning of the experiment occurred by establishing the sampling grid with 89 points regularly spaced in 50 x 50 meters). May be better to be along the lines (Initially sampling grids with 89 points regularly spaced in 50 x 50 meters were established) (lines 67-68).
Response 6: Thanks for pointing this out. We agreed and added the requested information to the paper.
Comment 7: obtained by measuring the ears with a graduated ruler; NFE, NGF (line 137). These abbreviations are not correct as not mentioned before.
Response 7: Thanks for pointing this out. We agreed and added the requested information to the paper.
Reviewer 2 Report
Comments and Suggestions for Authors
The authors conducted an interesting study on analysing spatial variability of apparent soil electrical conductivity in zones grown with maize at different densities. The authors used soil electrical conductivity as one of the parameter to indicate the behaviour of maize crop. The results are promising, however, the manuscript needs more details and justification as follows:
1. Authors need to elaborate, what kind of "behaviour" they like to observe in maize. Please see abstract.
2. Authors need to mention clearly previous studies related to measurement of electrical conductivity in literature. Even advanced studies on electrical conductivity and unsaturated soil moisture dynamics have been recently published (Relationship between bioelectricity and soil–water characteristics of biochar-aided plant microbial fuel cell).
3. Introduction needs to be expanded considering point no. 2.
4. Page 3, Line 100: The soil moisture was determined using very traditional approach. Why not measure volumetric water content instead of gravimetric. Authors can refer to measurement of soil water matric potential and water content based on "Effect of Three Different Types of Biochar on Bioelectricity Generated from Plant Microbial Fuel Cells under Unsaturated Soil Condition".
5. Limitations of measurement of electrical cond. sensor and also its procedure need to be highlighted.
6. Can authors conduct such study for a long term (1 -2 years) to consider climate change effects.
7. What are limitations of GIS method used in the study.
8. Though, Table 1 gives statistical results, however, authors should keep in mind that value of EC and soil moisture are highly inter related and dynamic with respect to time (of measurement), depth of soil and boundary condition (rainfall or evaporation). It is hard to conclude in general just by looking at basic statistics of the data. A time varying plots of these parameters is more useful.
9. Authors need to explain scientific mechanism that leads to different observations in MZ1 and MZ 2. Please refer to figure 5.
Author Response
Comment 1: Authors need to elaborate, what kind of "behaviour" they like to observe in maize. Please see abstract.
Response 1: Thanks for pointing this out. We agreed and added the requested information to the paper.
Comment 2: Authors need to mention clearly previous studies related to measurement of electrical conductivity in literature. Even advanced studies on electrical conductivity and unsaturated soil moisture dynamics have been recently published (Relationship between bioelectricity and soil–water characteristics of biochar-aided plant microbial fuel cell).
Response 2: Thanks for pointing this out. We respect your suggestion but we understand that, given the limitation in paper size, the works cited are sufficient to elucidate the problem presented. In total, 10 papers related to ECa were cited.
Comment 3: Introduction needs to be expanded considering point no. 2.
Response 3: Thanks for pointing this out. We respect your suggestion but we understand that, given the limitation in paper size, the works cited are sufficient to elucidate the problem presented. In total, 10 papers related to ECa were cited.
Comment 4: Page 3, Line 100: The soil moisture was determined using very traditional approach. Why not measure volumetric water content instead of gravimetric. Authors can refer to measurement of soil water matric potential and water content based on "Effect of Three Different Types of Biochar on Bioelectricity Generated from Plant Microbial Fuel Cells under Unsaturated Soil Condition".
Response 4: Thanks for pointing this out. When measuring soil apparent electrical conductivity, we usually quantify soil moisture in volumetric form, as the interest in knowing water content is associated with the relationship between water and soil electrical conductivity, as there is a positive correlation between these variables. For this reason, we quantify the water content in the soil in volumetric form.
Comment 5: Limitations of measurement of electrical cond. sensor and also its procedure need to be highlighted.
Response 5: Thanks for pointing this out. We agreed and added the requested information to the paper.
Comment 6: Can authors conduct such study for a long term (1 -2 years) to consider climate change effects.
Response 6: Thanks for pointing this out. This suggestion is valid and we intend to carry out a similar study in La Niña and El Niño years, but with soybean cultivation, which has greater added value for Brazilian farmers.
Comment 7: What are limitations of GIS method used in the study.
Response 7: Thanks for pointing this out. Basically, the limitations are associated with the operational knowledge of the software, its plugins and file types. It is also necessary for the user to have knowledge of statistical and geostatistical techniques to model spatial variability and creating reliable maps.
Comment 8: Though, Table 1 gives statistical results, however, authors should keep in mind that value of EC and soil moisture are highly inter related and dynamic with respect to time (of measurement), depth of soil and boundary condition (rainfall or evaporation). It is hard to conclude in general just by looking at basic statistics of the data. A time varying plots of these parameters is more useful.
Response 8: Thanks for pointing this out. We agree with your point. Our intention was to present to the reader the characterization of soil moisture at the time we carried out the ECa mapping. As mentioned, there is a high correlation between humidity and ECa. In practice, wetter locations have a higher ECa value, even in conditions of low soil humidity this behavior is maintained, making mapping in the field possible.
Comment 9: Authors need to explain scientific mechanism that leads to different observations in MZ1 and MZ 2. Please refer to figure 5.
Response 9: Thanks for pointing this out. We agreed and added the requested information to the paper.
Round 2
Reviewer 1 Report
Comments and Suggestions for Authors
No further comments, as all queries have been adequately addressed
Author Response
We thank you for the valuable contributions proposed by the Reviewer.Reviewer 2 Report
Comments and Suggestions for Authors
I give minor revisions again. Authors should consider more relevant studies in introduction and also discussion of results.
Comments on the Quality of English LanguageI give minor revisions again. Authors should consider more relevant studies in introduction and also discussion of results.
Author Response
Comments 1: I give minor revisions again. Authors should consider more relevant studies in introduction and also discussion of results.
Response 1: Thanks for pointing this out. We inform you that the following bibliographies have been added (Introduction, Discussion):
Sibonginkosi, N.; Mzwandile, M.; Tamado, T. Effect of plant density on growth and yield of maize [Zea Mays (l.)] hybrids at luyengo, middleveld of Eswatini. Asi. Plant Res. J., v. 3(3-4), p. 1-9, 2019. DOI: https://doi.org/10.9734/APRJ/2019/v3i3-430066
Yuan, Y.; Shi, B.; Yost, R.; Liu, X.; Tian, Y.; Zhu, Y.; Cao, W.; Cao, Q. Optimization of management zone delineation for precision crop management in an intensive farming system. Plants, v. 11, 2611, 2022. DOI: https://doi.org/10.3390/plants11192611
Corwin, D.L.; Lesch, S.M. Apparent soil electrical conductivity measurements in agriculture. Comput Electron. Agr., v.46, n.1-3, p. 11-43, 2005. DOI: https://doi.org/10.1016/j.compag.2004.10.005
Gelain, E.; Bottega, E.L.; Motomiya, A.V. de A.; Oliveira, Z.B. de. Variabilidade espacial e correlação dos atributos do solo com produtividade do milho e da soja. Nativa, v. 9, n. 5, p. 536-543, 2021. DOI: https://doi.org/10.31413/nativa.v9i5.11717
Monteiro, A.B.; Stöcker, C.M. Espaçamento entrelinhas de semeadura e produtividade da cultura do milho irrigado por aspersão. Rev. Bras. de Meio Ambi., v.8, n.4., p. 111-121, 2020.
Oliveira, F. de A.; Silva, J.C. da; Santos, D.P. dos; Barreto, J.A.S.; Silva, C.B. da; Santos, M.A.L. dos; Santos, V.R. dos. Increasing levels of irrigation and higher plant density increase the yield of green corn. J. of Develop., v.6, n.7, p. 43371-43381, 2020. DOI: http://dx.doi.org/10.34117/bjdv6n7-088
Zhang, L.; Zhang, Z.; Luo, Y.; Cao, J.; Li, Z. Optimizing genotype-environment-management interactions for maize farmers to adapt to climate change in different agro-ecological zones across China. Sci. Total Environ., 728, 138614, 2020. DOI: https://doi.org/10.1016/j.scitotenv.2020.138614